# Towards a Neurosymbolic Reasoning System Grounded in Schematic Representations

**François Olivier**  OLIVIER@CRIL.FR  and  **Zied Bouraoui**  BOURAOUI@CRIL.FR

*CRIL CNRS & Artois University, Lens, France*

**Editors:** Leilani H. Gilpin, Eleonora Giunchiglia, Pascal Hitzler, and Emile van Krieken

## Abstract

Despite significant progress in natural language understanding, Large Language Models (LLMs) remain error-prone when performing logical reasoning, often lacking the robust mental representations that enable human-like comprehension. We introduce a prototype neurosymbolic system, Embodied-LM, that grounds understanding and logical reasoning in schematic representations based on image schemas—recurring patterns derived from sensorimotor experience that structure human cognition. Our system operationalizes the spatial foundations of these cognitive structures using declarative spatial reasoning within Answer Set Programming. Through evaluation on logical deduction problems, we demonstrate that LLMs can be guided to interpret scenarios through embodied cognitive structures, that these structures can be formalized as executable programs, and that the resulting representations support effective logical reasoning with enhanced interpretability. While our current implementation focuses on spatial primitives, it establishes the computational foundation for incorporating more complex and dynamic representations.

## 1. Introduction

Despite significant progress in natural language understanding, Large Language Models (LLMs) continue to show considerable difficulties when performing simulations of described situations, such as tracking objects moved between containers (Tamari et al., 2020; McCoy et al., 2023; Mahowald et al., 2024), or reasoning soundly on problems with many premises (Callewaert et al., 2025). These limitations often stem from their lack of robust mental representations, preventing them from comprehending situations in ways comparable to humans.

Cognitive research revealed that humans understand natural language by relying on a finite set of conceptual primitives (Table 1) that they combine and utilize in mental simulation processes (Johnson, 1987; Mandler and Cánovas, 2014; Hedblom et al., 2024). These conceptual primitives are derived from early sensorimotor experiences and can be ranked by complexity into spatial, spatiotemporal, and force-dynamic categories, with the latter involving fully abstract notions such as force application. When combined, these primitives form what are called *image schemas*—the recurring patterns that gave this cognitive theory its name.

We introduce Embodied-LM, a proof-of-concept neurosymbolic system that provides a first computational realization of an image schema-based reasoning framework (Olivier and Bouraoui, 2025). Its architecture consists in leveraging LLMs' interpretive capabilities to identify appropriate schematic structures for given scenarios, and then translate these into formal programs processed by Clingo (Gebser et al., 2016) enhanced with the Declarative

Table 1: Classification of conceptual primitives from (Hedblom et al., 2024). The notion UMPH corresponds to the application of a force.

|  | entity | relational | attributive |
|---|---|---|---|
| **spatial** | OBJECT
CONTAINER
PATH
REGION
DOWN (/UP) | LOCATION
START_PATH
END_PATH
CONTACT
CONTAINED
SMALLER(/LARGER)
PART_OF | OPEN
CLOSED
EMPTY
OCCUPIED
FULL |
| **spatio-temporal** |  | PERMANENCE | MOTION
AT_REST
ANIMATE_MOTION
INANIMATE_MOTION |
| **force dynamic** |  | LINK | active-UMPH
passive-UMPH |

Spatial Reasoning (DSR) framework (Bhatt et al., 2011) to enable spatial reasoning capabilities. While the theoretical approach envisions a complete system incorporating spatial, spatiotemporal, and force-dynamic primitives (Olivier and Bouraoui, 2025), this proof-of-concept implementation focuses on the foundational spatial primitives, demonstrating that even this subset enables effective reasoning across multiple tasks with performance comparable to state-of-the-art models. Our work not only shows how systems grounded in human cognitive structures can effectively perform logical reasoning tasks, but also provides a computational foundation to build upon for more sophisticated and dynamic reasoning capabilities.

## 2. Related Work

Current neurosymbolic approaches for logical reasoning usually follow a common architectural pattern where LLMs generate formal representations subsequently processed by symbolic solvers. Logic-LM (Pan et al., 2023) employs a multi-formalism approach, translating natural language into first-order logic, constraint satisfaction problems, or SAT encodings, then delegating reasoning to specialized solvers. In contrast, VERUS-LM (Callewaert et al., 2025) adopts a unified approach, using a single rich formalism (FO($\cdot$)) to handle diverse reasoning tasks while introducing improved prompting mechanisms and knowledge-query separation to enhance efficiency and reusability. Logic-LM++ (Kirtania et al., 2024) incorporates pairwise comparison techniques to evaluate symbolic formulations, addressing cases where syntactically correct formulations remain semantically inadequate. Ishay et al. (Ishay et al., 2023) demonstrate that LLMs can generate complex Answer Set Programming representations from natural language descriptions of logic puzzles, though the generated programs often contain errors requiring human correction.

The formalization of image schemas has been explored through several approaches in previous work (Frank and Raubal, 1999), with qualitative calculi emerging as a prominent choice in recent research. Notably, Hedblom's work (Hedblom, 2020) made significant progress by combining various calculi such as Region Connection Calculus and Qualitative Trajectory Calculus with Linear Temporal Logic to represent both spatial and temporal dimensions of image schemas. Recently, Hedblom et al. proposed the Diagrammatic Image Schema Language (DISL) (Hedblom et al., 2024), a systematic diagrammatic representation for image schemas.

Within machine learning communities, Wachowiak et al. have explored how artificial agents can capture implicit human intuitions underlying natural language (Wicke and Wachowiak, 2024), introducing systematic methods for classifying natural language expressions according to underlying image schema structures (Wachowiak and Gromann, 2022). This work demonstrates growing recognition that image schemas provide important organizational principles for understanding human-AI interaction and language comprehension.

Regarding the embedding of the Declarative Spatial Reasoning framework (Bhatt et al., 2011) in Answer Set Programming, we can cite ASPMT(QS) (Wałęga et al., 2017), ASP(ST) (Schultz et al., 2018), and more recently Clingo2DSR (Li and Schultz, 2024). Our symbolic module differs from these systems by providing a fully declarative framework specifically designed for modeling schematic structures that underlie natural language comprehension. Early work on implementing image schemas within declarative programming can be found in (Suchan et al., 2015).

## 3. Reasoning Through Embodied Cognition

The field of embodied cognition emerged as a fundamental challenge to classical cognitive science by the end of the 20th century, demonstrating that our minds are not isolated symbol-processing computers but inextricably linked to bodily experiences. This became particularly evident in how we understand and use language, as Lakoff and Johnson's groundbreaking work in 'Metaphors We Live By' (Lakoff and Johnson, 1980) demonstrated by showing that we comprehend abstract concepts (the target domain) by relying on our physical experiences as a source domain—we understand time through location ("the future is ahead of us"), importance through size ("this is a big deal"), emotions through spatial orientation ("I'm feeling down"), or states through containment ("being in trouble").

To bridge the gap between bodily experience and thought, Johnson (Johnson, 1987) introduced image schemas—recurring patterns abstracted from our sensorimotor interactions—and showed their pervasive role in structuring human thought across both concrete and abstract domains. These schemas have received robust experimental confirmation across multiple studies (Richardson et al., 2001; Mandler and Cánovas, 2014) and have proven fruitful even in non-linguistic domains such as mathematics (Lakoff and Núñez, 2000). For instance, the image schema of OBJECT_INTO_CONTAINER, which arises from our early physical experiences of putting objects into containers (e.g., cups and buckets), later serves as a source domain for understanding literal sentences like "Bill is in the house", more abstract ones such as "Berlin is in Germany" or "to be in love", and mathematical expressions such as "$2 \in \mathbb{N}$". Over the years, researchers have decomposed image schemas into even more fundamental constituents (Table 1) to provide finer-grained explanations of their

compositional structure (Mandler and Cánovas, 2014; Hedblom et al., 2024). For instance, GOING_IN can be described through the combination of OBJECT, CONTAINER, and PATH primitives.

The key insight of image schemas lies in their productive capacity. While a primitive like PATH exists at an abstract level—serving as a template for understanding any sequential, directional, or ordering situation—it can generate specific spatial configurations that make implicit relationships explicit. Consider the sentence "Alice is older than Bill, and Charles is younger than Bill." To understand and reason about this sentence, image schema theory claims that we activate the PATH schema to create a timeline on which we place the different people so that temporal precedence maps to spatial positioning (Figure 1).

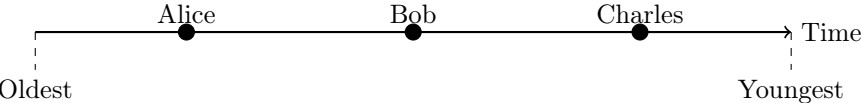

Figure 1: Temporal relationships naturally map to spatial positions on a timeline, illustrating how humans use spatial schemas to understand abstract concepts.

By placing these individuals along the timeline according to their relative ages, this spatial arrangement immediately reveals additional temporal relationships that weren't explicitly stated—such as Alice being older than Charles. Shimojima termed such inference a "free ride" in diagrammatic reasoning (Shimojima, 2015; Olivier, 2022), as spatial arrangements embed logical constraints in their geometric structure, making implicit relationships immediately apparent without any calculations. Our system, Embodied-LM, operationalizes this principle computationally.

## 4. Implementation

Our approach is designed for scenarios that present a *context* describing a situation, a *question* about that context and possible answer *options*. We focus on reasoning tasks within consistent contexts, though the approach can easily be extended to handle consistency checking tasks. Embodied-LM leverages LLMs to interpret scenarios through schematic structures that capture the underlying logical constraints and produce corresponding input programs for an Answer Set Programming system enhanced with the Declarative Spatial Reasoning framework, therefore creating executable representations that enable systematic logical inference.[1]

### 4.1. Formalizing Image Schemas in the DSR framework

The Declarative Spatial Reasoning (DSR) framework (Bhatt et al., 2011) enables users to declare arbitrary qualitative spatial relations and reason about the constraints implied among them. It relies primarily on techniques of analytic geometry by defining objects through parameters and relations through equations and inequalities involving these parameters (Preparata and Shamos, 2012).

---

1. Code and data available at https://github.com/fcs-olivier/embodied-lm.

The spatial entities in Table 1 correspond to geometric shapes that can be characterized as objects in the DSR framework and formalized through parametric functions: a simple OBJECT entity corresponds to a point defined by two parameters ($x$ and $y$ coordinates), a PATH can be formalized as a line segment defined by a start and end point ($x^s, y^s$ and $x^e, y^e$), a CONTAINER corresponds to any object with an interior—for instance, a circle defined by a center point plus a radius parameter, or a rectangle defined by bottom-left and top-right points.

The spatial relational primitives from Table 1 correspond to spatial relations that can be defined in the DSR framework through equations and inequalities. Figure 2 illustrates such formalization by detailing the *inside* relation between a point and a rectangle, capturing the intuitive notion of containment.

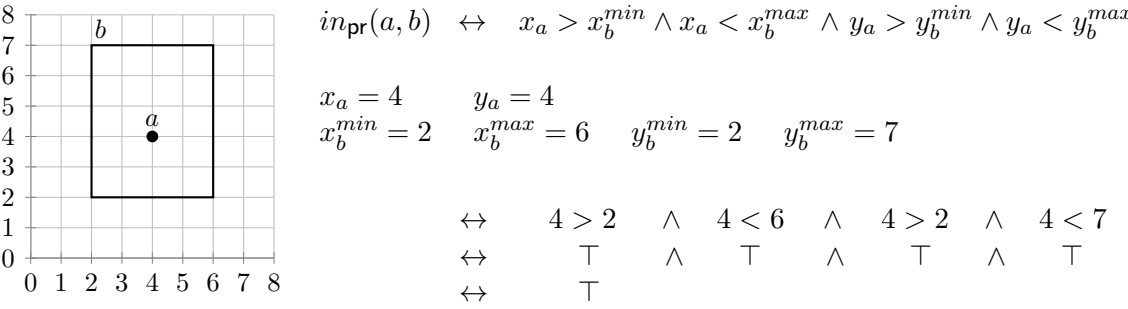

Figure 2: Spatial relationships are formalized in the DSR framework through coordinate constraints. Point $a$ is inside rectangle $b$ when its coordinates fall strictly within the rectangle's boundaries on both dimensions.

Table 2 presents some of the spatial relations used in the present paper and their formal definitions. The suffix indicates predicate types: pp for 'point-point', pr for 'point-rectangle', ps for 'point-segment' and so on. All the spatial relational primitives can be characterized in this way, with any higher abstraction level being achieved by defining unions of geometric sorts (e.g., defining CONTACT between any shapes). Once used within a declarative programming language as detailed in what follows, these relations enable the definition of spatial attributive primitives and more complex spatial configurations.

Table 2: Spatial relations and their logical definitions.

| Relation | Logical Definition |
|---|---|
| $samePlace_{\mathsf{pp}}(a,b)$ | $x_a = x_b \wedge y_a = y_b$ |
| $left_{\mathsf{pp}}(a,b)$ | $x_a < x_b$ |
| $right_{\mathsf{pp}}(a,b)$ | $x_a > x_b$ |
| $below_{\mathsf{pp}}(a,b)$ | $y_a > y_b$ |
| $in_{\mathsf{pr}}(a,b)$ | $x_a > x_b^{min} \wedge y_a > y_b^{min} \wedge x_a < x_b^{max} \wedge y_a < y_b^{max}$ |
| $left_{\mathsf{rr}}(a,b)$ | $x_a^{max} \leq x_b^{min} \wedge y_a^{max} > y_b^{min} \wedge y_a^{min} < y_b^{max}$ |
| $overlap_{\mathsf{rr}}(a,b)$ | $x_a^{min} < x_b^{max} \wedge y_a^{min} < y_b^{max} \wedge x_a^{max} > x_b^{min} \wedge y_a^{max} > y_b^{min}$ |
| $on_{\mathsf{ps}}(a,b)$ | $(x_a - x_b^s) \cdot (y_b^e - y_b^s) - (y_a - y_b^s) \cdot (x_b^e - x_b^s) = 0$ |
| | $\wedge \min(x_b^s, x_b^e) \leq x_a \leq \max(x_b^s, x_b^e) \wedge \min(y_b^s, y_b^e) \leq y_a \leq \max(y_b^s, y_b^e)$ |

## 4.2. Embedding Declarative Spatial Reasoning in Answer Set Programming

Answer Set Programming (ASP) provides an ideal foundation for implementing image schemas due to its declarative nature and non-monotonic reasoning capabilities that will be useful later on for implementing temporal and dynamic primitives. The unordered declaration property of ASP enables natural compositional reasoning where multiple schematic structures can be combined without concerns about procedural ordering—essential for handling real-world problems that require multiple interacting image schemas.

We extend standard ASP syntax (Calimeri et al., 2020; Kaminski et al., 2023) to include spatial theory atoms that state spatial relationships between entities. Spatial atoms are similar to those presented in Table 2, having the general form $p_{\mathsf{g}_1,...,\mathsf{g}_n}(entity_1, ..., entity_n)$ where $p$ is the relation name, $\mathsf{g}_1, ..., \mathsf{g}_n$ denote the geometric sorts of the entities, and $entity_1, ..., entity_n$ are the spatial objects involved in the relation. The geometry of each spatial object is declared as a fact within the program.

The semantics of extended programs can be defined within the logic of Here-and-There with constraints (Cabalar et al., 2020), where spatial predicates are interpreted through denotations that map each constraint atom to the set of coordinate valuations satisfying the corresponding geometric definitions. For instance, $left_{\mathsf{pp}}(a, b)$ is satisfied when $x_a < x_b$ in the coordinate assignment. Intuitively, the semantics relies on a two-level structure: at the logical level, standard ASP rules determine which spatial relationships must hold based on problem constraints, while at the spatial level, the constraints determine whether these required relationships can be geometrically realized through coordinate assignments. A stable model is found when the spatial relationships required by the ASP rules can be geometrically realized through valid coordinate assignments.

The integration mechanism employs theory propagation between the ASP solver and Z3 SMT solver through a custom propagator that maintains consistency between logical and spatial constraints. When spatial atoms are set to true during the solving process, the propagator tests whether adding each spatial atom would make the constraint system inconsistent—if so, it adds nogood clauses. For atoms that maintain consistency, they are asserted into the spatial constraint system, and the propagator examines free literals to determine if any spatial relationships are either implied by or inconsistent with the current spatial configuration, adding appropriate clauses or nogoods to guide the search toward spatially realizable solutions.

## 4.3. LLM-Guided Schema Generation

The LLM component (GPT-4) of our system generates these ASP programs through prompting that guides interpretation of natural language scenarios using image schemas. A general system prompt (see Appendix A) informs the LLM about available spatial predicates, their intended semantics, and the syntactic requirements. A prompt for formalizing the context of scenarios (see Appendix B), although currently still specific to the problem type, provides example patterns for encoding spatial relationships, enabling the LLM to generalize these patterns to new problem instances. To ensure reliable program generation, the system includes syntax error detection and satisfiability checking, asking the LLM to produce a new input program in case of syntax errors or unsatisfiable constraints up to three attempts (after which the scenario is discarded if still unsuccessful).

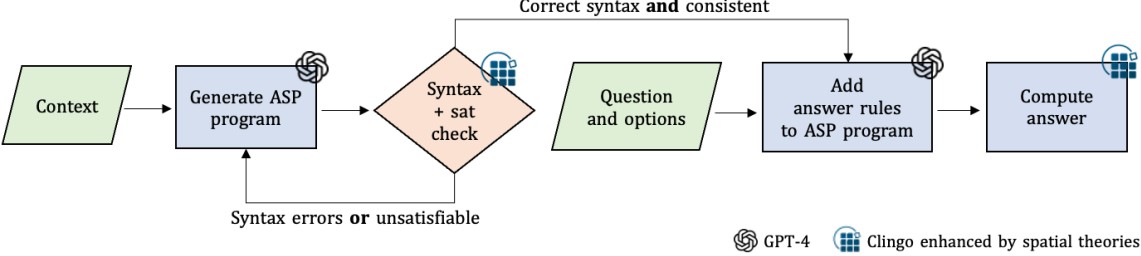

Figure 3: Architecture of Embodied-LM.

Once a valid ASP program representing the context of the scenario is generated, GPT-4 is called a second time (see Appendix C) to add the question and options to the program in accordance with the context formalization. Each option becomes an additional rule, called an *answer rule*, whose purpose is to succeed when the corresponding answer option is true. We instruct GPT-4 to use a predicate *answer*/1 in the heads of such rules, where the argument identifies the corresponding option, and leverages Clingo `#show` directives on this predicate (combined with the `--project=show` option) to output only the answers. Complete input programs can be seen in Appendix D. By additionally using the Clingo option `--enum-mode=cautious`, which computes the cautious consequences (atoms true in all stable models), we ensure that only options validated across all stable models are considered as answers to the question. A witness for each answer can be generated using the assignment provided by Z3 for the parametric variables, enabling visualization of the spatial configurations that satisfy the logical constraints. The overall architecture of our approach is depicted in Figure 3.

## 5. Experimental Validation

To validate our framework's core principles, we applied Embodied-LM to different problems requiring logical reasoning. Rather than optimizing for benchmark performance, our primary objective was to demonstrate some fundamental capabilities: first, that LLMs can be guided to interpret problems through embodied cognitive structures; second, that these structures can be formalized as executable programs; and third, that the resulting representations enable effective logical reasoning while maintaining interpretability. We focus on reasoning tasks that activate the PATH and CONTAINER primitives.

### 5.1. Reasoning with PATH: LogicalDeduction Dataset

We evaluated our approach on the LogicalDeduction dataset (Srivastava et al., 2023), which presents multiple-choice questions requiring deductive reasoning about ordered objects. These problems naturally invoke schematic reasoning because they involve positioning entities along various dimensions such as temporal sequences, spatial arrangements, or ordinal rankings. Figure 4 illustrates a representative problem from the dataset. The specific prompts used to introduce the context and the question-options to the LLM are provided in Appendix B.1 and C.1, respectively.

---

**Context:** The following paragraphs each describe a set of five objects arranged in a fixed order. The statements are logically consistent within each paragraph.
In an antique car show, there are five vehicles: a truck, a motorcycle, a limousine, a station wagon, and a sedan.
The limousine is older than the truck.
The sedan is newer than the motorcycle.
The station wagon is the oldest.
The limousine is newer than the sedan.
**Question:** Which of the following is true?
**Options:** A) The truck is the second-oldest.
     B) The motorcycle is the second-oldest.
     C) The limousine is the second-oldest.
     D) The station wagon is the second-oldest.
     E) The sedan is the second-oldest.

---

Figure 4: Representative problem from LogicalDeduction dataset (Srivastava et al., 2023) naturally invoking a schematic representation with the PATH primitive.

When presented with such an example, GPT-4 positions the different entities from the context on a line representing the PATH primitive (using the $on_{\mathsf{ps}}$ predicate) and interprets older/newer relationships through corresponding left/right or below/above spatial positioning, leading to Clingo identifying option B as the correct answer. An example of input programs generated by the LLM for this scenario is presented in Appendix D.1. When tested on the entire dataset, our system achieved 91% accuracy, as shown in Figure 6 along with comparisons to other approaches on the same dataset (Callewaert et al., 2025).

### 5.2. Reasoning with CONTAINER: Zebra Puzzles

For more complex reasoning scenarios, we examined an instance of a Zebra puzzle involving three houses with different colors and various attributes distributed among them (Figure 5). These problems naturally invoke the CONTAINER image schema by mapping diverse relationships—living in, drinking, owning—to a single image schema of spatial containment as a source domain. The reasoning process then requires determining which entities belong within which bounded spaces while maintaining spatial relationships between the containers themselves. The prompts used to introduce the context and question-options to the LLM are provided in Appendices B.2 and C.2, respectively.

A generated ASP program is presented in Appendix D.2, where all problem constraints are encoded using only three core spatial predicates—$in_{\mathsf{pr}}$, $left_{\mathsf{rr}}$ and $right_{\mathsf{rr}}$—along with commonsense knowledge such as non-overlapping houses and object sort classifications. The system finds two possible models that differ only in whether the dog or fox lives with the Swiss versus the Cuban, but in both models the zebra is consistently owned by the Greek, therefore validating option C as the correct answer. Figure 6 displays witnesses for both models, showing the spatial configurations that satisfy all constraints.

---

**Context:** There are three houses painted in different colors: blue, red, and green.
TEA is drunk in the RED house.
The CUBAN drinks MILK.
The SWISS man lives in the first house on the left.
The GREEK lives in a house to the right of the CUBAN's house.
BEERS are drunk in a house to the right of the FOX's owner.
The DOG's owner lives to the left of the house where the GREEK lives.
The ZEBRA's owner lives in the BLUE house.
**Question:** What nationality is the zebra's owner?
**Options:** A) The zebra's owner is Cuban.
            B) The zebra's owner is Swiss.
            C) The zebra's owner is Greek.

---

Figure 5: Instance of a zebra puzzle.

| System | Score |
|--------|-------|
| SymbCoT | 93.00 |
| **Our Approach** | **91.00** |
| VERUS-LM | 88.67 |
| Logic-LM | 87.63 |
| GPT4-CoT | 75.25 |
| GPT4 | 71.33 |

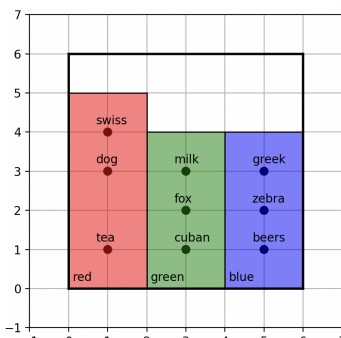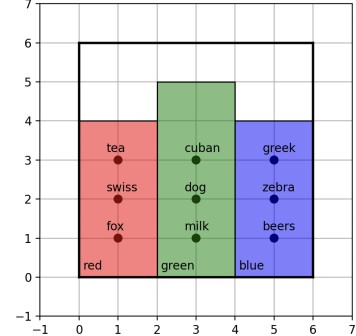

Figure 6: Left: Performance comparison on the LogicalDeduction dataset. Results for other systems from (Callewaert et al., 2025). Right: Witnesses of the two possible models for the Zebra problem.

## 6. Results and Discussion

Our experimental results achieve the primary goal of demonstrating that AI systems can reason using schematic structures in ways that parallel human cognitive processes. Importantly, we demonstrate that a single image schema can ground multiple types of information across different problem domains—the PATH schema maps temporal sequences, spatial arrangements, and ordinal rankings to positional relations, while the CONTAINER schema unifies diverse relationships like ownership through spatial containment. This demonstrates that a universal schematic language can ground diverse scenarios, constituting a promising foundation for extending the approach to any dataset where these image schemas can be detected, while minimizing risks encountered by other neurosymbolic systems relying on classical logic in two important ways: (i) the predicate names remain consistent across problems, and (ii) more fundamentally, the meaning of spatial predicates is objective and predefined, alleviating the burden on LLMs to correctly constrain predicate meanings within input programs.

Beyond this conceptual validation, our approach maintains competitive performance with 91% accuracy on the LogicalDeduction dataset compared to other neurosymbolic approaches (Figure 6), although prompting remains quite specific to the task at hand. The Zebra puzzle resolution demonstrates similar capacity to handle complex reasoning scenarios, with the additional advantage that our system can systematically evaluate multiple possible models through Clingo's cautious reasoning mode. This capability potentially addresses a limitation where LLMs would likely struggle to consider all valid interpretations and may fail to recognize that alternative models can serve as counterexamples that invalidate certain answer options.

The framework also provides enhanced interpretability compared to black-box neural approaches. Since the ASP program generated by the LLM is directly available for inspection, errors in mapping or comprehension can be diagnosed and potentially corrected through prompt refinement. Specifically, this interpretability enables insightful error analysis on the LogicalDeduction dataset, where we found that the LLM sometimes fails to conserve the same interpretation along the reasoning process (e.g., establishing that "newest" corresponds to leftmost positioning but later introducing constraints assuming rightward positioning). This suggests opportunities for improvement through additional spatial relations, such as $startsAt_{\mathsf{sp}}$ for the PATH schema, or prompting that could enforce directional consistency. Psychological research comparing human and system-generated schematic representations might provide further solutions to these mapping challenges while offering interesting cognitive insights.

Finally, several technical directions could enhance the system's capabilities. The neural component presents opportunities for more sophisticated prompting strategies, including separating interpretation and program generation into distinct phases. Additionally, a problem-specific prompt selection function could be designed based on the identified image schemas underlying each problem's resolution. A dedicated syntactic checker could also replace Clingo's error detection to improve efficiency, though the checker's design matters little since we have observed that GPT-4 more often produces correct programs when starting fresh rather than attempting to correct existing ones.

## 7. Conclusion

We presented Embodied-LM, a prototype neurosymbolic system that grounds logical reasoning in schematic representations based on image schemas. Through evaluation on logical deduction and puzzle-solving tasks, our approach provides conceptual validation that spatial cognitive structures can be formalized as executable programs, enabling systematic logical inference while maintaining interpretability.

Even if our current implementation focuses on spatial primitives, it establishes the computational foundation for processing more complex schematic structures. Building on this groundwork, the framework opens pathways for incorporating more diverse reasoning scenarios including mathematical, syllogistic, and propositional logic problems, while future extensions that integrate spatio-temporal and force dynamic primitives could enable resolving classic AI problems such as Tower of Hanoi, river-crossing puzzles, or Blocks World scenarios.

## Acknowledgments

This work was supported by the French National Research Agency (ANR) under grant ANR-22-CE23-0002 ERIANA.

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

## Appendix A. System Prompt

**System Prompt**

You are an expert at writing declarative programs that encode the interpretation of natural language narratives through image schemas - recurring patterns derived from sensorimotor experience that structure human cognition. Declarative programs are written in Answer Set Programming augmented with spatial theory predicates.
It is important that the schematic representation enables reasoning on the logical constraints stated in the narratives.

OBJECT DECLARATION:
Objects in narratives correspond to geometric objects in the program. Specify the geometric sort of one or more objects (using pooling) by "point(obj1;obj2;...)."

BUILTIN SPATIAL PREDICATES AVAILABLE:
The suffixes "_ps", "_pp", "_p, _rr" indicate the relation type (e.g., _ps is for point-segment) and arity (e.g., leftmost_p is of arity 1):
- on_ps(P1,S2): Point P1 is on segment S2
- samePlace_pp(P1,P2): Points P1 and P2 are at the same location
- left_pp(P1,P2) / right_pp(P1,P2): Point P1 is to the left/right of point P2
- leftmost_p(P) / rightmost_p(P): there is no point to the left/right of P
- above_pp(P1,P2) / below_pp(P1,P2): Point P1 is above/below point P2
- uppermost_p(P) / lowermost_p(P): there is no point above/below P
- overlap_rr(R1,R2): Rectangle R1 overlaps with rectangle R2
- in_pr(P1,R2): Point P1 is inside rectangle R2
- left_rr(R1,R2) / right_rr(R1,R2): Rectangle R1 is to the left/right of rectangle R2

SYNTAX RULES:
1. Use :- ... for integrity constraints that must be satisfied.
2. Anonymous variables (_) cannot be used in the head of rules but only in the body. For instance, avoid "left_pp(tractor,_)." as a fact.
3. Variables must always be safe: they must appear in a positive literal in the rule body to ensure it is properly bound.

GENERAL REQUIREMENTS:
1. Always maintain coherence in your spatial interpretation. For instance, if you interpret a concept like "newer" as "to the right" in one rule,
you must consistently use this same interpretation for writing the other rules.
2. Output only the raw ASP code with no formatting. DO NOT use triple backticks (''')
or markdown code blocks.
3. Comment any non-ASP content with % symbols.

## Appendix B. Context Formalization Prompts

### B.1. LogicalDeduction Dataset

**Context Prompt**

Generate the Clingo program with the qualitative spatial predicates available to encode the spatial schematic interpretation of the natural language input.

EXAMPLE RULES FOR NARRATIVE INFORMATION:

```
% In an antique car show, there are five vehicles: a convertible, a sedan, a
tractor, a minivan, and a limousine.
% Declare the objects and their geometric sort
point(convertible; sedan; tractor; minivan; limousine).
% Align all car points on a timeline to constrain possible solutions
on_ps(P,timeline) :- point(P).
:- samePlace_pp(_,_).   % Disallow any points to be at the same place

% The apples are less expensive than the mangoes
left_pp(apple, mango).

% The kiwis are the second-cheapest (note how the variable for the counted items takes
the first argument)
:- #count{X : left_pp(X, kiwi)} != 1.

% The van is the second newest
:- #count{X : right_pp(X, van)} != 1.

% The tractor is the newest
rightmost_p(tractor).

{CONTEXT}
```

## B.2. Zebra Puzzle

```
Context Prompt

Generate a Clingo-based program for solving zebra puzzle problems through image schemas.

%%%%%%%%%% SORTS DECLARATION %%%%%%%%%%
pet(fox;dog).      % each object is associated with a classical sort
person(cuban;greek).
house(blue;green).
...

item(X):- pet(X).
item(X):- person(X).
...

point(X) :- item(X). % classical sorts are mapped to geometric sorts
rect(X) :- house(X).

isA(Pet, pet):- pet(Pet).
isA(Person, person):- person(Person).
...

%%%%%%%%%% CONSTRAINTS %%%%%%%%%%
% Houses are apart from each other
:- overlap_rr(_,_).

% Every item is contained in a house
:- not in_pr(Item,_), item(Item).
```

```
% A house only contains one person who owns one pet
:- in_pr(Item1,House), in_pr(Item2,House), Item1!=Item2, isA(Item1, Sort1),
isA(Item2, Sort2), Sort1=Sort2.

% The GREEK lives in a house to the right of the CUBAN.
right_rr(House1, House2):- in_pr(greek, House1), in_pr(cuban,House2),
house(House1), house(House2).

% The CUBAN lives in the first house on the left.
in_pr(cuban,House):- not left_rr(_,House), house(House).

{CONTEXT}
```

# Appendix C. Question and Options Formalization Prompts

## C.1. LogicalDeduction Dataset

### Question and Options Prompt

```
Transform the natural language question into Clingo-based rules that will make the
answer available in the system's output. These rules will extend the existing ASP
input program.

EXAMPLES OF OPTIONS AND THEIR MODELING:
answer(a):- #count{X : left_pp(X,cardinal)}=2. % A) The cardinal is the third
                                                            from the left
answer(b):- rightmost_p(apple).          % B) The apple is the most expensive
answer(c):- left_pp(tractor, minivan).   % C) the tractor is older than the minivan
answer(d) :- #count{X : right_pp(X, white)} = 1. % D) The white book is the second
                                                            from the right

REQUIRED OUTPUT:
- Generate ASP rules that model each answer option, so that I can simply append these
rules to the INPUT PROGRAM.
- All constants must be lowercase (e.g., 'answer(a)', do not write 'answer(A)').
- Add a "#show answer/1" so that only the predicate answer is shown in the output.

{CONTEXT}
{QUESTIONS AND OPTIONS}
{INPUT PROGRAM}
```

## C.2. Zebra Puzzle

### Question and Options Prompt

```
Transform the natural language question into Clingo-based rules that will make the
answer available in the system's output. These rules will extend the existing ASP
input program.

EXAMPLES OF OPTIONS AND THEIR MODELING:
answer(a):- in_pr(zebra, House), in_pr(cuban, House). % A) The zebra's owner is Cuban
```

```
REQUIRED OUTPUT:
- Generate ASP rules that model each answer option, so that I can simply append these
  rules to the INPUT PROGRAM.
- All constants must be lowercase (e.g., 'answer(a)', do not write 'answer(A)').
- Add a "#show answer/1" so that only the predicate answer is shown in the output.

{CONTEXT}
{QUESTIONS AND OPTIONS}
{INPUT PROGRAM}
```

# Appendix D.  Examples of Generated Programs

## D.1.  LogicalDeduction Dataset

### Example of Generated ASP Program

```
point(truck; motorcycle; limousine; station_wagon; sedan).

% All vehicles are aligned to reflect an order based on 'newer' or 'older'
on_ps(V, vehicle_line) :- point(V).

% Ensure no two vehicles share the same position
:- samePlace_pp(_, _).

% The limousine is older than the truck
left_pp(limousine, truck).

% The sedan is newer than the motorcycle
right_pp(sedan, motorcycle).

% The station wagon is the oldest
leftmost_p(station_wagon).

% The limousine is newer than the sedan
right_pp(limousine, sedan).

answer(a) :- #count{X : left_pp(X, truck)} = 1.
answer(b) :- #count{X : left_pp(X, motorcycle)} = 1.
answer(c) :- #count{X : left_pp(X, limousine)} = 1.
answer(d) :- #count{X : left_pp(X, station_wagon)} = 1.
answer(e) :- #count{X : left_pp(X, sedan)} = 1.

#show answer/1.
```

## D.2. Zebra Puzzle

**Example of Generated ASP Program**

```
% Define objects and their classical sorts
pet(fox;dog;zebra).
person(cuban;greek;swiss).
house(blue;red;green).
drink(tea;milk;beer).

% Define items and assign items to classical sorts
item(X):- pet(X).
item(X):- person(X).
item(X):- drink(X).

% Map classical sorts to geometric sorts
point(X) :- item(X).
rect(X) :- house(X).

% Define associations for items and sorts
isA(Pet, pet):- pet(Pet).
isA(Person, person):- person(Person).
isA(Drink, drink):- drink(Drink).

%%%%%%%%%%% CONSTRAINTS %%%%%%%%%%%%
% Houses are apart from each other
:- overlap_rr(_,_).

% Every item is contained in a house
:- not in_pr(Item,_), item(Item).

% A house only contains one person, one pet, and one drink
:- in_pr(Item1,House), in_pr(Item2,House), Item1!=Item2, isA(Item1, Sort1),
isA(Item2, Sort2), Sort1=Sort2.

% The GREEK lives in a house to the right of the CUBAN's house
right_rr(House1, House2):- in_pr(greek, House1), in_pr(cuban, House2),
house(House1), house(House2).

% The CUBAN drinks MILK
in_pr(milk, House):- in_pr(cuban, House), house(House).

% TEA is drunk in the RED house
in_pr(tea, red).

% The SWISS lives in the first house on the left
in_pr(swiss, House):- not left_rr(_, House), house(House).

% BEERS are drunk in a house to the right of the FOX's owner
right_rr(House1, House2):- in_pr(beer, House1), in_pr(fox, House2),
house(House1), house(House2).

% The DOG's owner lives to the left of the house where the GREEK lives
left_rr(House1, House2):- in_pr(dog, House1), in_pr(greek, House2),
house(House1), house(House2).
```

```
% The ZEBRA's owner lives in the BLUE house
in_pr(zebra, blue).

% Define answer options based on the narrative's question
answer(a):- in_pr(zebra, House), in_pr(cuban, House).
answer(b):- in_pr(zebra, House), in_pr(swiss, House).
answer(c):- in_pr(zebra, House), in_pr(greek, House).

% Show only the answer in the output
#show answer/1.
```

