# OpenReview forum: "Towards a Neurosymbolic Reasoning System Grounded in Schematic Representations"
_nesyconf.org/NeSy/2025/Conference_Phase_2 — NeSy 2025 - Phase 2 Poster_

### Official Review · Reviewer_FiaY · 2025-07-07
**Interesting work based on usng image schemas to ground neuro-symbolic reasoning**

**Rating:** 7
**Confidence:** 3

**Review:**

Novelty: The performance table show solid performance but not optimal when compared to SOTA. The contribution seams simplistic.

Abstract: Well written and clear in motivation and contribution and impact. The deviation from the SOTA could be pointed out clearer.

Text: Is well written.

Overall presentation: Figure 3 is low resolution (pixelish) and has too small font sizes, the caption is only describing the obvious instead of providing a clear explanatory main message of what the reader should get form this image.

**Anonymity:**

Remain anonymous

---

### Official Review · Reviewer_h8tR · 2025-07-08
**A Novel and Interpretable Framework for Spatial Reasoning with LLMs**

**Rating:** 6
**Confidence:** 2

**Review:**

[Summary] They authors propose Embodied-LM, a NeSy framework that integrates large language models (LLMs) with formal spatial reasoning via image schemas and logic programming. LLMs are guided to interpret problems using embodied cognitive structures—specifically, image schemas like path and container—which are then translated into spatial configurations and executable programs using the Declarative Spatial Reasoning framework and Clingo (Answer Set Programming).

[Review] This paper introduces a novel and computationally effective approach that combines LLMs with formal spatial reasoning via image schemas. Grounded in ideas from cognitive theory, the framework is both innovative and interpretable, as it produces symbolic representations through ASP that can be inspected and debugged.

However, there are several areas where the paper could be improved.

- References to the supplementary material would be clearer if they pointed to specific figures or sections (e.g., “Fig. S2” or “Section A.3”) rather than using vague phrases.
- The evaluation on the LogicalDeduction dataset lacks context about the baseline systems (e.g., Logic-LM, GPT-4-CoT); a brief explanation of these approaches and why they were chosen for comparison would help clarify and strengthen the performance claims.
- The paper mentions checking for syntax errors from the LLM up to three times, but it is unclear what happens if errors persist after the third attempt—whether a fallback strategy or manual intervention is applied.
- Adding the date when the GPT-4 was being used for the experiments, could improve the paper.
- The manuscript exceeds the venue’s 10-page limit and should be shortened accordingly.

Typo:
- Remove “*sample footnote” on the first page.

**Anonymity:**

Remain anonymous